# Gut microbiome differences among Mexican Americans with and without type 2 diabetes mellitus

**Amanda K. Kitten**[1,2], **Laurajo Ryan**[1,2], **Grace C. Lee**[1,2], **Bertha E. Flores**[3,4], **Kelly R. Reveles**[1,2,4] *

**1** College of Pharmacy, The University of Texas at Austin, Austin, Texas, United States of America, **2** Pharmacotherapy Education and Research Center, University of Texas Health San Antonio, San Antonio, Texas, United States of America, **3** School of Nursing, University of Texas Health San Antonio, San Antonio, Texas, United States of America, **4** Research to Advance Community Health Center, University of Texas Health San Antonio, San Antonio, Texas, United States of America

\* kdaniels46@utexas.edu

## Abstract

### Purpose

Type 2 diabetes mellitus (T2DM) is an urgent public health problem and disproportionately affects Mexican Americans. The gut microbiome contributes to the pathophysiology of diabetes; however, no studies have examined this association in Mexican-Americans. The objective of this study was to compare gut microbiome composition between Mexican-Americans with and without T2DM.

### Methods

This was a cross-sectional study of volunteers from San Antonio, TX. Subjects were 18 years or older and self-identified as Mexican American. Subjects were grouped by prior T2DM diagnosis. Eligible subjects attended a clinic visit to provide demographic and medical information. Thereafter, subjects recorded their dietary intake for three days and collected a stool sample on the fourth day. Stool 16s rRNA sequences were classified into operational taxonomic units (OTUs) via the mothur bayesian classifier and referenced to the Green-genes database. Shannon diversity and bacterial taxa relative abundance were compared between groups using the Wilcoxon rank sum test. Beta diversity was estimated using Bray-Curtis indices and compared between groups using PERMANOVA.

### Results

Thirty-seven subjects were included, 14 (38%) with diabetes and 23 (62%) without diabetes. Groups were well-matched by body mass index and comorbid conditions. Shannon diversity was not significantly different between those with and without T2DM (3.26 vs. 3.31; p = 0.341). Beta diversity was not significantly associated with T2DM diagnosis (p = 0.201). The relative abundance of the most common bacterial phyla and families did not significantly differ between groups; however, 16 OTUs were significantly different between groups.

**Data Availability Statement:** All files are available from the NCBI Sequence Read Archive (BioProject ID PRJNA719138).

**Funding:** This study was funded by the Research to Advance Community Health Center, University of

Texas San Antonio (reach.uthscsa.edu) awarded to KRR. The funders had no role in study design, data collection and analysis, decision to publish, or preparation of the manuscript. Research support was provided by the FIRST Outpatient Research Unit, Institute for Integration of Medicine & Science/Clinical & Translational Science Award, UT Health San Antonio (UL1 TR002645).

**Competing interests:** The authors have declared that no competing interests exist.

## Conclusions

Although alpha diversity was not significantly different between diabetic and non-diabetic Mexican Americans, the abundance of certain bacterial taxa were significantly different between groups.

## Introduction

The human microbiome is a rapidly expanding area of clinical research. Scientific literature examining the role of the microbiome in human health has increased substantially over the last ten years [1]. This has led to the identification of the microbiome as a major contributor to human health and disease [2]. In particular, studies have established a connection between the gut microbiome and the pathophysiology of type 2 diabetes mellitus (T2DM). Specifically, the gut microbiome plays a role in the mediation of inflammation, metabolite production, gut mucosa integrity, and metabolic hormone secretion [3, 4].

Several large studies have compared cohorts of human subjects with and without T2DM to quantify compositional differences and have identified several gut microbial profiles that are thought to contribute to the risk of developing diabetes. For example, studies demonstrated a depletion of several bacterial taxa, including the Bifidobacterium genus, Firmicutes phylum, and Roseburia genus in patients with T2DM [5–10]. Studies have consistently demonstrated enrichment of the Lactobacillus genus in subjects with T2DM [5–7, 9, 10]. Lactobacillus has been implicated in obesity and is thought to be an immune-modulating bacteria [11, 12]. Finally, reduced bacterial diversity was associated with T2DM in one study [13].

Hispanic subjects experience a higher risk of developing T2DM compared to non-Hispanic Whites, likely due to genetics and possibly dietary differences [14, 15]. Researchers have also identified differences in the gut microbiome composition of Hispanics compared the general United States population; Ross et al. compared the gut microbial composition between Hispanics in South Texas and the Human Microbiome Project and found significant taxonomical differences between the two groups [16]. Notably, the differences seen between this Hispanic cohort, 87.3% of whom did not have T2DM, and the overall Human Microbiome Project population were similar to the differences seen in studies that compared control subjects to subjects with T2DM. Additionally, a review of studies investigating the gut microbiome's role in the development of obesity and diabetes in Latin American subjects and in the Hispanic/Latino community living in the United States concluded that a relationship likely exists [17]. Given this information, further investigation of the gut microbiome as a potential predisposing factor for T2DM in the Hispanic population is needed. Additionally, no prior studies discuss how baseline characteristics other than T2DM status might have contributed to findings regarding gut microbiome composition.

## Materials and methods

### Study design

This was a cross-sectional study of volunteers from San Antonio, TX and surrounding areas from June 2017 to July 2018. This study was approved by the Institutional Review Board at UT Health San Antonio. Subjects were recruited using newspaper advertisements in the *San Antonio Express-News*, *Southside Reporter*, and *Conexión*. Flyers were also placed in the Medical Arts and Research Center (MARC) in the South Texas Medical Center. Due to this

recruitment method, the sampled population most likely represents relatively healthy community-dwelling individuals and may not be representative of all T2DM and non-diabetic patient populations. Those interested in participating called the research team, and a research team member pre-screened subjects using a detailed questionnaire. If participants successfully completed pre-screening, they were scheduled for a research visit at the MARC.

## Study population

Subjects were eligible for inclusion if they were at least 18 years old and self-identified as Mexican American. Subjects were excluded if they had a history of prior gastrointestinal surgery that altered the anatomy of the gastrointestinal tract. Medication use that warranted exclusion included (1) chronic daily use of any drug meant to alter gastrointestinal secretory or motor function (e.g., prokinetic agents, narcotic analgesics, laxatives, anticholinergics, anti-diarrheals) or (2) any use of antibiotics, gastric-acid suppressing medications, or probiotics in the previous two months. Subjects were divided into groups based on T2DM status. Subjects were considered to have T2DM if they had been previously diagnosed with T2DM and were currently receiving active treatment for diabetes.

## Data collection

Subjects attended a single research visit. After the subjects provided written informed consent, they were asked to complete a demographic and health questionnaire. A complete list of data collected can be found in S1 Appendix. Subjects were then provided with a three-day food diary and stool sample collection kit to take home for the remainder of the study procedures. Subjects were instructed to complete the food diary for the three days following the study visit. They were instructed to document all dietary intake (food and beverage) and the approximate quantity they consumed each day. Total caloric intake was estimated using the United States Department of Agriculture Food Composition Databases [18]. The study team used these data to calculate healthy eating index (HEI) scores for participants. Points were assigned based on the Update of the Healthy Eating Index HEI-2015 scoring system, which assesses how closely food intake aligns with the Dietary Guidelines for Americans (DGA); the higher the score, the better the alignment with DGA [19]. Finally, participants were instructed to collect a single stool sample on day four using the DNA Genotek OMNIgene Gut Kit provided at their initial visit. After collecting the stool sample, they shipped their sample to the investigators in a pre-paid envelope.

## Stool sample collection, processing, and sequencing

Stool samples were stored at -80 degrees Celsius until sequencing. Microbiome Insights, Inc. performed DNA extraction, sequencing, and analysis. DNA was extracted from specimens using the MoBIO PowerMag Soil DNA Isolation Bead Plate and KingFisher™ robot. Bacterial 16S rRNA genes were PCR-amplified using primers targeting the V4 region. Primers were comprised of Illumina adapters, an 8-nucleotide index sequence, a 10-nucleotide pad sequence to prevent hairpin formation, and a gene-specific primer. Amplicons were sequenced using the Illumina MiSeq 300-bp paired-end kit (v.3). Taxonomic classifications were denoised and classified using the Greengenes v. 13_8 database, and clustered into 97%-similarity operational taxonomic units (OTUs) using the mothur software package (v. 1.39.5). Raw sequences and metadata have been deposited into the NCBI Sequence Read Archive (BioProject ID PRJNA719138).

### Data and statistical analyses

Baseline characteristics were compared between the group with T2DM and non-diabetes groups using JMP 14.0.0® (SAS Institute, Cary, NC, USA) (S1 Appendix). The Wilcoxon rank sum test was used to compare non-parametric continuous data, whereas the chi-square test (or Fisher's Exact test as appropriate) was used for nominal data. P-values<0.05 indicated statistical significance.

Alpha diversity was estimated with the Shannon index on raw OTU abundance tables after filtering out contaminants. OTUs were excluded if they occurred in fewer than 10% of samples and with a count of less than three. Shannon diversity was compared between groups using the Wilcoxon rank sum test. Beta diversity was measured using Bray-Curtis indices and visualized using Principal Coordinates Analysis (PCoA). Variation in community structure was assessed with permutational multivariate analyses of variance (PERMANOVA) with treatment group as the main fixed factor and using 4,999 permutations for significance testing. The impact of age and obesity on beta diversity was evaluated using PERMANOVA.

## Results

The study included 37 subjects, 14 (38%) with T2DM and 23 (62%) without diabetes. The median age (IQR) was 59 years (48–68), and 27 (73%) were female (Table 1). Overall, participants were overweight, and about half (46%) had hypertension. Subjects with diabetes were older than those without diabetes (68 versus 55 years; p = 0.003). Prevalence of disease of the cardiovascular system (e.g. hypertension, dyslipidemia, and history of myocardial infarction [MI]) were not significantly different between groups; however, rates were numerically higher in those with T2DM. Twelve of the 14 subjects with diabetes (86%) took metformin.

There was no significant difference in median Shannon diversity between subjects with (3.26) and without (3.31) T2DM (p = 0.341) (Fig 1). The most dominant phyla for both groups were Bacteroidetes and Firmicutes. Though not reaching statistical significance, the T2DM group had a higher percent abundance of Firmicutes (36.5% vs. 30.0%, p = 0.131), Proteobacteria (6.4% vs. 5.7%, p = 0.485), and Verrucomicrobia (5.3% vs. 2.5%, p = 0.062) and a lower abundance of Bacteroidetes (49.2% vs. 59.2%, p = 0.131) compared to the non-T2DM group (Fig 2). At the family level, the largest numeric differences between the T2DM and non-T2DM groups were with the Bacteroidaceae (28.8% vs. 21.6%, p = 0.399) and Prevotellaceae (13.1% vs. 30.2%, p = 0.172) though neither reached statistical significance (Fig 3). There was a significant difference in the relative abundance of 16 OTUs between groups. Fig 4 displays the percent relative abundance of the eight most abundant of these OTUs; p<0.001 for all comparisons between T2DM and non-T2DM groups.

There were no significant differences in beta diversity between subjects with and without T2DM as measured by PERMANOVA (p = 0.201) (Fig 5). Neither age (p = 0.196) nor obesity (p = 0.120) were significantly associated with beta diversity. Clustering was not significantly different after accounting for HEI score (p = 0.496), diabetes status (p = 0.347), and metformin use (p = 0.767).

## Discussion

This is one of the first studies to compare gut microbiome composition between Mexican Americans with and without T2DM. We collected demographic and other health-related information that could potentially contribute to microbiome differences between groups. We compared two well-matched groups; both groups had high median BMIs and similar rates of cardiovascular comorbidities. Our study did not find a significant difference in alpha diversity between groups; however, we did note significant differences in 16 bacterial taxa.

**Table 1. Baseline characteristics.**

| Characteristic | All subjects (n = 37) | Diabetes (n = 14) | No diabetes (n = 23) | p-value |
|---|---|---|---|---|
| Age, median (IQR), years | 59 (48–68) | 68 (59–72) | 55 (38–61) | 0.003 |
| Female, no. (%) | 27 (73) | 9 (64) | 18 (78) | 0.454 |
| BMI [a], median (IQR), kg/m$^2$ | 28.7 (26.6–34) | 30 (26–36) | 28 (27–31) | 0.465 |
| Metformin, no. (%) | 12 (33) | 12(86) | 0 (0) | ≤0.001 |
| Sulfonylurea, no. (%) | 3 (8) | 3 (21) | 0 (0) | 0.012 |
| GLP-1 RA, no. (%) | 3 (8) | 3 (21) | 0 (0) | 0.012 |
| Insulin, no. (%) | 2 (5) | 2 (14) | 0 (0) | 0.044 |
| HMG-CoA reductase inhibitor | 10 (28) | 8 (57) | 2 (9) | 0.002 |
| ACEI/ARB | 11 (32) | 8 (57) | 3 (15) | 0.009 |
| Beta-blocker | 5 (15) | 3 (21) | 2 (10) | 0.358 |
| Diuretic | 2 (5) | 0 (0) | 2 (14) | 0.044 |
| Highest level of education, no. (%) | | | | 0.289 |
| High school or equivalent | 8 (22) | 2 (14) | 6 (26) | |
| Some college, no degree | 14 (38) | 8 (57) | 6 (26) | |
| Associate degree | 6 (16) | 3 (21) | 3 (13) | |
| Bachelor's degree | 4 (11) | 0 (0) | 4 (17) | |
| Master's degree | 1 (3) | 0(0) | 1 (4) | |
| Employment status, no (%) | | | | ≤0.001 |
| Retired | 15 (41) | 11(79) | 4 (17) | |
| Employed for wages | 17 (46) | 1 (7) | 16 (70) | |
| Out of work/looking for work | 5 (14) | 2 (14) | 3 (13) | |
| Hypertension, no. (%) | 17 (46) | 9 (53) | 5 (25) | 0.079 |
| Dyslipidemia, no. (%) | 10 (27) | 5 (36) | 5 (22) | 0.357 |
| History of MI, no. (%) | 1 (3) | 1 (7) | 0 (0) | 0.378 |
| History of cancer, no. (%) | 1 (3) | 1 (7) | 0 (0) | 0.378 |
| Depression, no. (%) | 1 (3) | 1 (7) | 0 (0) | 0.378 |
| IBS, no. (%) | 1 (3) | 0 (0) | 1 (4) | 1.000 |
| HEI score, median (IQR) | 53.5 (42.7–66.6) | 62.0 (60.0–65.8) | 48.5 (40.0–68.4) | 0.234 |
| Household income ($), median (IQR) | 24,000 (4,850–55,000) | 25,500 (1,275–56,250) | 24,000 (8,400–60,000) | 0.742 |
| Mexico birth, no. (%) | | | | |
| Subjects | 3 (8) | 2 (14) | 1 (4) | 0.290 |
| Parents | 9 (24) | 3 (21) | 6 (26) | 0.747 |
| Grandparents | 22 (59) | 8 (57) | 14 (61) | 0.823 |

IQR, interquartile range; BMI, body mass index; GLP-1 RA, glucagon-like peptide-1 receptor agonist; ACEI, angiotensin converting enzyme inhibitor; ARB, angiotensin II receptor blocker; MI, myocardial infarction; IBS, irritable bowel syndrome; HEI, healthy eating index scores

[a]BMI not reported by one subject

Although alpha diversity has been shown to be a marker of multiple diseases, including obesity [20], colorectal cancer [21], and type 1 diabetes [22], multiple gut microbiome studies in T2DM demonstrated that alpha diversity is not significantly different between well-matched subjects with and without T2DM [5, 7]. One study that included 15 (31%) Hispanic patients found that alpha diversity was slightly, but non-significantly lower in patients with prediabetes (5.26) or T2DM (5.21) compared to those without diabetes [8]. However, one study by Zhang et al. determined that alpha diversity was negatively correlated with insulin resistance [13]. It is unclear why alpha diversity results are so variable. Multiple factors can contribute to gut microbiome diversity, including diet and medications [23]. Though not significantly different,

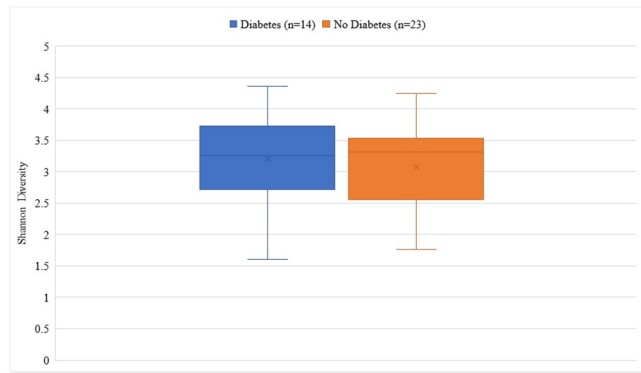

**Fig 1. Shannon diversity by diabetes status.**

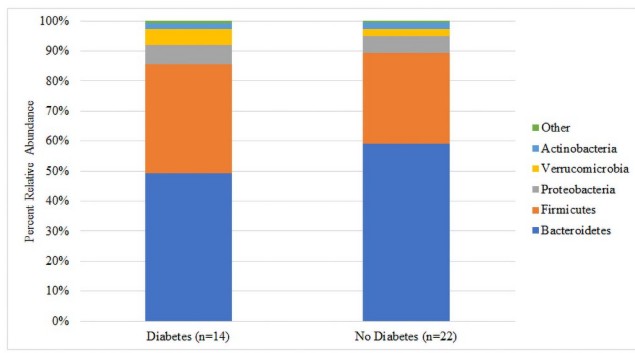

**Fig 2. Percent relative abundance of bacterial phyla by diabetes status.**

subjects with T2DM had a numerically higher HEI score, indicating overall superior diet. It is difficult to interpret the clinical significance of this 13.5-point difference in the context of our study because previous studies used HEI scores to stratify subjects, while our study treated HEI scores as a confounding variable. It is possible that differences in HEI scores contributed to greater than expected bacterial diversity in T2DM subjects. Interestingly, the national average HEI score is 59, which is closer to the scores of our T2DM subjects compared to those without T2DM [24].

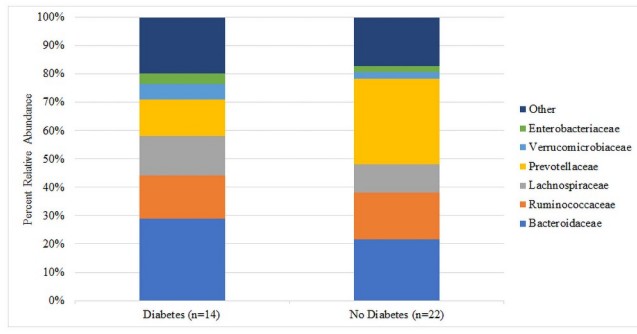

**Fig 3. Percent relative abundance of bacterial families by diabetes status.**

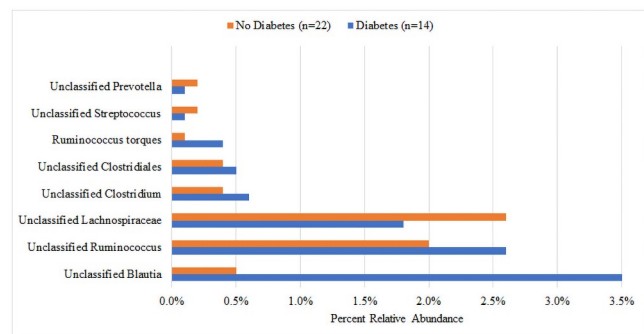

**Fig 4. Percent relative abundance of the most abundant significantly different OTUs.**

Another major difference that could have contributed to lack of a difference in alpha diversity between groups is medication usage, specifically metformin, which is considered first-line therapy for T2DM according to the American Diabetes Association guidelines [25]. Metformin use has been shown to increase gut microbial diversity compared to subjects with T2DM not taking metformin [26]. The aforementioned gut microbiome studies did not report subjects' medication use; therefore, we were not able to determine whether metformin use contributed to the lack of differences seen in alpha diversity between groups [5–8, 10, 13]. In this study, 86% of subjects with T2DM took metformin compared to none in the non-T2DM group. This high rate of metformin use by those with T2DM could have led to increased alpha diversity in that group, resulting in similar alpha diversities between groups.

The microbial composition did not differ between T2DM and non-T2DM groups at the phylum and family levels, but significant differences were noted in 16 OTUs. previous studies have evaluated the abundance of various bacterial taxa. Bacteria significantly depleted in subjects with T2DM include: *Bifidobacterium* genus [5, 6], Firmicutes phylum [7, 8], and *Roseburia* genus [7, 9, 10]. The relationship between these microbes and metabolism have been described previously. In mice studies, an increase in gut Bifidobacterium attributable to prebiotic fiber ingestion resulted in improved glucose tolerance and decreases in inflammatory markers [27]. A high relative abundance of Firmicutes, especially in relation to Bacteroidetes,

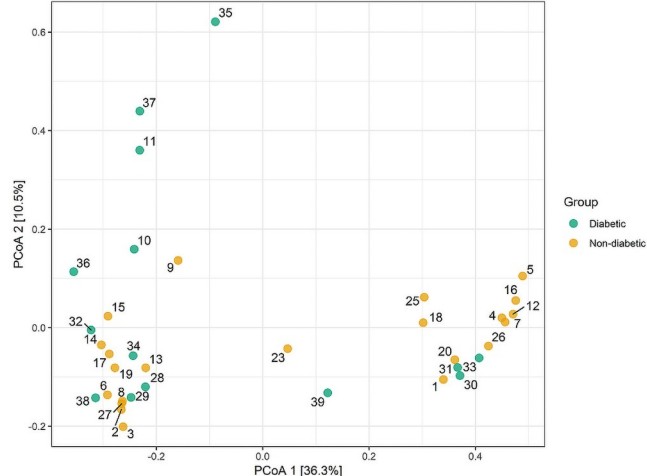

**Fig 5. Principal coordinates analysis of microbiome beta diversity by diabetes status.**

has been implicated in obesity and high BMI [28]. Interestingly, the Firmicutes phylum contains many SCFA-producing bacteria that confer metabolic benefits [10]. Our study found no significant differences in the relative abundance of *Lactobacillus*, *Bacteroides*, *Prevotella*, Clostridia, *Bifidobacterium*, or *Roseburia*., possibly because our study was not powered to detect a difference.

The *Streptococcus* genus was significantly more abundant in non-T2DM subjects compared to subjects with T2DM. *Streptococcus* has been associated with atherosclerotic cardiovascular disease, hypertension, and enhanced thrombotic risk [29, 30]. It is therefore surprising that *Streptococcus* was enriched in our non-T2DM subjects as they had overall lower rates of cardiovascular-related diseases, and T2DM itself is a risk factor for cardiovascular disease. One possible explanation for this is the modulatory effects of metformin on the microbiome, which many of the T2DM subjects (86%) reported taking. Differences in diet also could have contributed to the observed difference in *Streptococcus* relative abundance.

Contrary to previous studies, there was no association between beta-diversity and T2DM status [5, 10]. This is likely because the control groups in previous studies were healthier in general, whereas the non-T2DM subjects in our study had high rates of cardiovascular-related disease and a high median BMI, making their overall health more similar to the T2DM group. It is also possible that the similar degrees of beta-diversity are due to the shared ethnicity of the two groups. Furthermore, as mentioned previously, metformin demonstrates microbiome-modulating effects in T2DM [26].

There were several limitations to the study which stem primarily from its design as a small, cross-sectional pilot study. For example, the study may not have been powered to detect the differences in microbial abundance identified in previous studies. However, despite this small sample size, we were able to identify several novel differences between groups. Another limitation was that all demographic and health information was self-reported by the subjects as opposed to being extracted from medical records. Up to one quarter of all people with diabetes are undiagnosed, so it is possible that some of our non-diabetic subjects have diabetes [14]. Furthermore, comorbid diseases, height, weight, and medication use could have been inaccurately reported. Finally, we could not control for all microbiome mediators.

## Conclusions

Although alpha diversity was not significantly different between Mexican Americans with and without T2DM, the abundance of certain bacterial taxa were significantly different between the groups. Several interesting findings from this study could stimulate further research on diabetes treatment and prevention. For example, the high relative abundance of *Streptococcus* in subjects without diabetes who also had lower rates of diseases that predispose for atherosclerotic cardiovascular disease warrants further investigation. Additionally, it is possible that metformin modulates multiple facets of metabolic health through its effects on the gut microbiome.

## Supporting information

**S1 Appendix. Clinical and microbial diversity and abundance data.**
(XLSX)

## Author Contributions

**Conceptualization:** Amanda K. Kitten, Laurajo Ryan, Grace C. Lee, Bertha E. Flores, Kelly R. Reveles.

Data curation: Amanda K. Kitten.

Formal analysis: Amanda K. Kitten, Kelly R. Reveles.

Funding acquisition: Kelly R. Reveles.

Investigation: Amanda K. Kitten.

Methodology: Amanda K. Kitten, Laurajo Ryan, Grace C. Lee, Bertha E. Flores, Kelly R. Reveles.

Project administration: Kelly R. Reveles.

Supervision: Laurajo Ryan, Grace C. Lee, Kelly R. Reveles.

Writing – original draft: Amanda K. Kitten.

Writing – review & editing: Laurajo Ryan, Grace C. Lee, Bertha E. Flores, Kelly R. Reveles.

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
