## [Decision Letter · Decision Letter 0]

23 Mar 2021

PONE-D-21-02685

Gut microbiome differences among Mexican Americans with and without type 2 diabetes mellitus

PLOS ONE

Dear Dr. Reveles,

Thank you for submitting your manuscript to PLOS ONE. After careful consideration, we feel that it has merit but does not fully meet PLOS ONE’s publication criteria as it currently stands. Therefore, we invite you to submit a revised version of the manuscript that addresses the points raised during the review process.

As presented, it is difficult to assess the soundness of the data, because of the confusing presentation. Please follow the Reviewer's advice to improve the data presentation.

We look forward to receiving your revised manuscript.

Kind regards,

Franck Carbonero, PhD

Academic Editor

PLOS ONE

Journal Requirements:

2. In your Methods section, please provide additional information about the participant recruitment method and the demographic details of your participants. Please ensure you have provided sufficient details to replicate the analyses such as:

a) the recruitment date range (month and year),

b) a description of any inclusion/exclusion criteria that were applied to participant recruitment,

c) a table of relevant demographic details,

d) a statement as to whether your sample can be considered representative of a larger population,

e) a description of how participants were recruited, and

f) descriptions of where participants were recruited and where the research took place.

3. We note that you are reporting an analysis of a microarray, next-generation sequencing, or deep sequencing data set. PLOS requires that authors comply with field-specific standards for preparation, recording, and deposition of data in repositories appropriate to their field. Please upload these data to a stable, public repository (such as ArrayExpress, Gene Expression Omnibus (GEO), DNA Data Bank of Japan (DDBJ), NCBI GenBank, NCBI Sequence Read Archive, or EMBL Nucleotide Sequence Database (ENA)). In your revised cover letter, please provide the relevant accession numbers that may be used to access these data. For a full list of recommended repositories, see http://journals.plos.org/plosone/s/data-availability#loc-omics or http://journals.plos.org/plosone/s/data-availability#loc-sequencing

4. Please amend the manuscript submission data (via Edit Submission) to include author Bertha E. Flores.

'..Research support was provided by the FIRST Outpatient Research Unit, Institute for Integration of Medicine & Science/Clinical & Translational Science Award, UT Health San Antonio (UL1 TR002645).'

'This study was funded by the Research to Advance Community Health Center, University of Texas San Antonio (reach.uthscsa.edu) awarded to KRR. The funders had no role in study design, data collection and analysis, decision to publish, or preparation of the manuscript.'

Additional Editor Comments:

The data needs to be presented in a more common fashion. I believe the authors are wrongly using the Wilcoxon as their data from two cohorts can't be paired. As suggested by Rewiewer 1, Table 2 need to be represented graphically with actual relative abundances. It is absolutely fine that there were no remarkable difference between groups. But the granular differences need to be shown with even more rigor. I would guess that some taxa (Haemophilus, Pyramidobacter) were extremely low abundant, and therefore may be irrelevant to present.

Reviewers' comments:

Reviewer's Responses to Questions

**Comments to the Author**

1. Is the manuscript technically sound, and do the data support the conclusions?

Reviewer #1: Partly

2. Has the statistical analysis been performed appropriately and rigorously? 

Reviewer #1: Yes

3. Have the authors made all data underlying the findings in their manuscript fully available?

Reviewer #1: No

4. Is the manuscript presented in an intelligible fashion and written in standard English?

Reviewer #1: Yes

5. Review Comments to the Author

Reviewer #1: Kitten and colleagues present a valuably and timely study of the gut microbiome of Mexican Americans with T2D. Indeed, T2D is a significant problem in this population and although we know from other populations that the microbiome plays an important role in T2D, there are almost no studies examining the microbiome Mexican Americans specifically. Though the sample size is quite small, this is nonetheless an important contribution to the literature. The manuscript is clear and well written, however there are some concerns that need to be addressed.

The microbiome data should be reported as relative abundance, not raw counts.

Table 2 is difficult to read, but this would probably be addressed by reporting values as relative abundance instead of counts. Also consider reporting the distributions graphically (as in Fig1) rather than as a table.

Table 2 mentions "normalized counts" but there is no discription of how this normalization was performed in the Methods section.

Will the authors make the microbiome data publicly available upon publication? It is common practice to upload all raw sequencing data the the NCBI SRA upon publication and cite the SRA Study Accession number in the mansucript.

6. PLOS authors have the option to publish the peer review history of their article (what does this mean?). If published, this will include your full peer review and any attached files.

Reviewer #1: No

---

## [Author Response · Author response to Decision Letter 0]

12 Apr 2021

Thank you for the thoughtful comments. Please see the attached response to reviewers document for our responses and corresponding manuscript changes.

---

## [Decision Letter · Decision Letter 1]

23 Apr 2021

Gut microbiome differences among Mexican Americans with and without type 2 diabetes mellitus

PONE-D-21-02685R1

Dear Dr. Reveles,

We’re pleased to inform you that your manuscript has been judged scientifically suitable for publication and will be formally accepted for publication once it meets all outstanding technical requirements.

Kind regards,

Franck Carbonero, PhD

Academic Editor

PLOS ONE

Additional Editor Comments (optional):

Reviewers' comments:

Reviewer's Responses to Questions

**Comments to the Author**

1. If the authors have adequately addressed your comments raised in a previous round of review and you feel that this manuscript is now acceptable for publication, you may indicate that here to bypass the “Comments to the Author” section, enter your conflict of interest statement in the “Confidential to Editor” section, and submit your "Accept" recommendation.

Reviewer #1: All comments have been addressed

2. Is the manuscript technically sound, and do the data support the conclusions?

Reviewer #1: Yes

3. Has the statistical analysis been performed appropriately and rigorously? 

Reviewer #1: Yes

4. Have the authors made all data underlying the findings in their manuscript fully available?

Reviewer #1: Yes

5. Is the manuscript presented in an intelligible fashion and written in standard English?

Reviewer #1: Yes

6. Review Comments to the Author

Reviewer #1: All of my concerns have been addressed in this revision.

7. PLOS authors have the option to publish the peer review history of their article (what does this mean?). If published, this will include your full peer review and any attached files.

Reviewer #1: **Yes: **Aleksandar David Kostic

---

## [Editor Report · Acceptance letter]

30 Apr 2021

PONE-D-21-02685R1 

Gut microbiome differences among Mexican Americans with and without type 2 diabetes mellitus 

Dear Dr. Reveles:

I'm pleased to inform you that your manuscript has been deemed suitable for publication in PLOS ONE. Congratulations! Your manuscript is now with our production department. 

Kind regards, 

on behalf of

Dr. Franck Carbonero 

Academic Editor

PLOS ONE